# Convergence of Pro-Stress and Pro-Inflammatory Signaling in the Central Noradrenergic System: Implications for Mood and Anxiety Disorders

Arthur Anthony A. Reyes and Daniel J. Chandler *

Department of Cell Biology and Neuroscience, Rowan-Virtua School of Osteopathic Medicine, Stratford, NJ 08084, USA; reyesa74@rowan.edu
* Correspondence: chandlerd@rowan.edu

**Abstract:** Mood and anxiety disorders are heterogeneous psychiatric diagnoses affecting millions. While the disease etiology is complex, various risk factors have been identified, such as stress. Stress is a neuroendocrine physiologic response to a stressor that promotes organism survival through adaptive processes and behavior. The central stress response, which drives behavioral and physiological change, is primarily mediated by activating the hypothalamic–pituitary–adrenal (HPA) axis. In addition to its effects on the HPA axis, stress activates the locus coeruleus (LC), a bilateral brainstem nucleus that projects broadly throughout the central nervous system and releases the catecholamine transmitter norepinephrine (NE). The combined activities of the LC–NE system and HPA axis work synergistically to produce timely adaptive physiological and behavioral responses to stress. While advantageous in the short term, chronic stress exposure can lead to HPA axis and LC dysregulation, which are thought to contribute to the etiology of several neuropsychiatric disease states. Notably, recent studies have also implicated neuroinflammation mediated by microglia as a risk factor in mood and anxiety disorders. Despite their combined association with mood and anxiety disorders, the potential links between stress and inflammation, and possible interactions between their respective signaling cascades, have not been well-explored. This brief review aims to summarize how LC is uniquely positioned to respond to both pro-stress and pro-inflammatory cues, and how their convergence in this site may contribute to the development of mood and anxiety disorders.

**Keywords:** stress; neuroinflammation; locus coeruleus; microglia; norepinephrine; mood disorders; anxiety disorders

## 1. Introduction

Mood and anxiety disorders are highly heterogeneous conditions encompassing multiple clinical diagnoses characterized by psychiatric and emotional disturbances that limit normal functioning in day-to-day life. Mood disorders are a group of mental disorders marked by emotional disruptions such as severe lows (depression), extreme highs (mania), or both [1], and include major depressive disorder (MDD) and bipolar disorder. Estimates from the National Comorbidity Survey Replication (NCS-R) suggest that mood disorders are highly prevalent, with an estimated 9.7% of the US adult population diagnosed and an additional 21.4% experiencing mood disorders at some time in their lives [2]. Anxiety disorders, on the other hand, are a group of mental disorders characterized by unfocused worry, pathological anxiety, and behavioral disturbances [3], and include generalized anxiety disorder, post-traumatic stress disorder (PTSD), and phobias [4]. Estimates suggest that 19.1% of adults in the US have been diagnosed with some form of anxiety disorder, with about 31% suffering from anxiety at some point in their life [2]. With the high prevalence of mood and anxiety disorders, it is no surprise that associated financial burdens are also high, with an annual estimated cost of $44 billion for mood disorders [5] and $42 billion for anxiety disorders [6]. On top of the tremendous financial burden, mood and anxiety

disorders also put an incalculable amount of pain and suffering on the diagnosed and their caregivers. In addition, mood and anxiety disorders are associated with an increased risk for cardiovascular disease [7], obesity [8], and alcohol-use disorder [9]. Despite the high prevalence and tremendous disease burden, treatment management for mood and anxiety disorders is inadequate, with many patients intractable to therapeutics, and many who do respond showing high rates of relapse.

Current management of mood and anxiety disorders involves psychotherapy, pharmacotherapy, or a combination of both. Pharmacotherapy in mood disorders uses antidepressants such as selective serotonin reuptake inhibitors (SSRIs) to reduce symptom severity. While widely used, antidepressant efficiency is inconsistent [10] and varies depending on symptom severity. Meta-analysis of randomized placebo-controlled trials reveals that, relative to placebo, patients with mild or moderate symptoms only experience slight benefits [11]. The addition of psychotherapy to pharmacotherapy, however, adds a slight benefit, resulting in a 66% reduction in depressive symptoms [12]. Likewise, combined therapy in anxiety disorders has only about 50% clinical efficacy, with 30% of adults relapsing 3–6 months after treatment discontinuation [3]. In addition to their less-than-optimal clinical efficiency, pharmacotherapy use is associated with adverse side effects, contributing to lower treatment compliance [13]. Together, these data highlight current mood and anxiety treatment inefficiencies. However, despite these deficiencies, the study of mood and anxiety disorders over the years has identified complex biological, social, and environmental risk factors common to both conditions.

Despite their distinct and complex clinical phenotypes, similar risk factors have been identified, such as genetics, sex, environment, and stress exposure. In both disorders, women show a higher diagnosis prevalence than men [14]. Barring social differences, such as coping styles and attitudes towards mental health, women still demonstrate a higher prevalence than males, suggesting a biological effect of sex in increasing the risk of mood or anxiety disorder diagnosis [15,16]. Similarly, a family member's mood or anxiety disorder diagnosis increases the likelihood of future diagnosis [17]. While genetics contribute to the increased likelihood of a future diagnosis, family history risk factors could include negative parenting styles and traumatic childhood experiences from living with a diagnosed caregiver [18,19]. Indeed, stress exposure through trauma is a well-studied risk factor in the emergence of mood and anxiety disorders [20,21]. Remarkably, a significant correlation exists between the onset of major depression and life-changing events in the previous three months [22]. More recently, studies have demonstrated the emergence of another biological risk factor: inflammation. Patients diagnosed with mood or anxiety disorders show increased levels of peripheral inflammation [23,24]. Together, these similarities provide clues as to the complex etiology of mood and anxiety disorders. While identifying the contributions of each risk factor is complex, considering both as reciprocally-interacting effectors of disease can enhance understanding of disease etiology and lead to better treatments. This review presents an integrated perspective wherein stress exposure, LC–NE dysfunction, and inflammation interact in mood and anxiety disorders. By emphasizing the complex reciprocal interactions between known risk factors in mood and anxiety disorders, we propose that LC represents a unique anatomic site whose integration of pro-stress and pro-inflammatory cues makes it uniquely positioned to contribute to the etiology of mood and anxiety disorders through neuronal and non-neuronal mechanisms.

## 2. The Stress Response and HPA Axis Dysregulation

Stress is an adaptive physiologic neuroendocrine response to a stressor that promotes organism survival [25]. Often referred to as the "fight-or-flight" response, stress promotes adaptive biological processes and behaviors that enable the organism to eliminate (fight) or avoid (flight) the stressor. In response to stress, external and internal cues are integrated at the level of the hypothalamus, triggering the release of corticotropin-releasing hormone (CRH) [26]. CRH travels to the pituitary gland located at the base of the brain. The pituitary gland produces adrenocorticotropin-releasing hormone (ACTH) in response to CRH, and

ACTH is released into the bloodstream through the hypophyseal portal system to reach its distant target, the adrenal cortex [27]. Within the adrenal cortex, ACTH binds with its receptors to produce glucocorticoids (GCs). GCs travel throughout the bloodstream to reach multiple target organs such as the liver, heart, and skeletal muscle, leading to increased gluconeogenesis, glucose mobilization, heart rate, and muscular contraction [25]. While the organ targets for corticosterone are vastly different in structure, function, and location, the net GC effect is to coordinate bodily functions to deal with the stressor properly.

Importantly, the activity of the HPA axis, like most biological cascades, is tightly regulated through a negative-feedback loop [28]. Within the HPA axis, corticosterone, the primary output effector molecule, signals to the hypothalamus and pituitary gland to downregulate the production of CRH and ACTH, respectively [29]. It is important to note that CRH and ACTH also provide negative feedback and thus decrease the production of downstream effector molecules. This negative-feedback regulation is necessary to maintain GC levels within a tight physiologic range and is crucial in preventing homeostatic disruption that can affect multiple organ systems [25]. However, despite the tight HPA axis control, various conditions, such as chronic illness, acute infections, and chronic stress exposure, can induce HPA axis dysregulation [30].

HPA axis dysregulation occurs when the HPA axis negative-feedback system response is blunted or completely unresponsive [31]. In a state of dysregulated HPA axis, homeostasis is disrupted, and the system fails to maintain physiologic levels of CRH, ACTH, and GCs. Non-physiologic levels of CRH, ACTH, or GCs have been implicated in various pathologies, including mood and anxiety disorders [25,32,33]. Indeed, HPA axis dysregulation was noted in patients with depression and anxiety [34] and is evident in patients with depression after human CRH (hCRH) challenge. Administration of hCRH typically decreases ACTH levels to a physiologic range. However, hCRH challenge in patients with depression and anxiety leads to a blunted response, suggesting a disruption of the normal HPA axis [31]. Similarly, clinical studies reveal that 35–65% of patients diagnosed with major depressive disorder (MDD) have higher plasma, salivary, and urinary GCs [34]. These studies demonstrate the effect of stress on HPA axis dysregulation. However, with its broad effects, chronic stress also affects different stress-responsive systems within the body, such as the LC–NE system.

## 3. The LC–NE System and Its Dysregulation under Chronic Stress

Locus coeruleus, a bilateral brainstem nucleus adjacent to the third ventricle, is the central nervous system's primary source of NE. Despite its small size, the approximately cylindrical 10,000-neuron nucleus in humans (3000 neurons in rodents) releases NE to different anatomically-distinct targets [35]. Well-studied targets of the LC–NE system include the prefrontal cortex, hippocampus, and amygdala, thus implicating the LC in associated cognitive and emotional functions such as decision-making, memory formation, and learning [36]. In addition to cortical targets, the LC also projects to other brainstem nuclei, implicating its role in the sleep–wake cycle and modulation of peripheral NE release [37,38]. LC's widespread projection and implication in various cognitive functions can be attributed to its heterogeneous nature. Current studies suggest that, despite the small size and widespread projections, LC anatomical and functional organization varies according to its efferent topography [35]. LC's heterogeneity allows the small nucleus to respond to signals from different brain regions while projecting to various targets [39]. LC is also characterized by protein kinase A-dependent spontaneous tonic firing [40]. Without modulatory neurotransmitters, LC discharges at a tonic firing rate of 0–5 Hz, which correlates to a basal NE release [41]. Interestingly, LC tonic firing correlates with different cognitive states. LC fires at a low tonic rate of 1–2 Hz in the awake state, which decreases during sleep [42]. In response to a stimulus or stressful events, however, LC firing increases to a higher tonic rate of 3–8 Hz [43]. Notably, during the stress response, CRF is released from the hypothalamus and several central brain nuclei onto LC, which promotes increased LC tonic discharge and anxiety-like behavior [40,43]. LC responses during stress

increase NE in critical brain targets, leading to increased alertness, memory consolidation, and learning [36]. Hence, the coordinated response between the HPA axis and the LC–NE system during stress provides the organism with a timely and well-orchestrated adaptive behavioral response. However, as with the HPA axis, the LC–NE system can also be dysregulated in response to chronic stress exposure.

It is thought that chronic stress-induced HPA axis dysregulation contributes to LC hyperactivity and dysregulation [44]. In animal models, acute stress exposure [45] and direct stimulation of LC or release of CRF within LC [43] have both been shown to increase LC activity and anxiety-like behavior. Similarly, LC chemogenetic inhibition after stressor exposure prevents anxiety-like behavior expression, suggesting a direct causal relationship between LC activity and anxiety-like behaviors [43]. Clinical studies similarly highlight a direct relationship between LC function and anxiety/negative effect: fMRI studies have shown increased LC neural responsiveness, along with behavioral and autonomic hyper-responsiveness, in PTSD patients in response to harmless stimuli [46]. Likewise, CSF NE levels are elevated in PTSD patients compared to control subjects. Moreover, the elevated CSF NE levels in PTSD patients positively correlated with scores on a clinician-administered PTSD scale [47]. Similar to PTSD patients, clinical studies demonstrate alterations in LC–NE system in MDD. While PTSD patients show increased NE, many clinical studies suggest a reduction in noradrenergic transmission in MDD [48]. Clinical studies point out that the reductions in noradrenergic transmission may be due to fewer LC neurons. Indeed, postmortem analysis of the brains of suicide victims with depression shows a 23% reduction in LC neurons and 38% lower density compared to normal subjects [49]. Reductions in noradrenergic transmission might also be due to alterations in auto-receptor functions. Furthermore, postmortem brain analysis of suicide victims with depression shows alterations in $\alpha$2 receptor functions. Compared to normal subjects without depression, suicide victims show increased $\alpha$2 auto-receptor binding with clonidine, an $\alpha$2 agonist, while no differences in $\alpha$1 agonist binding (yohimbine) were noted [50]. These results point to altered expression of $\alpha$2 auto-receptors in suicide patients. Likewise, radioligand binding assays reveal increased binding of $\alpha$2 agonist (clonidine) in suicide victims compared to control subjects [51]. Increased $\alpha$2 expression and binding suggest reductions in NE as $\alpha$2 agonist binding decreases NE release, consistent with clinical reports of reduced NE in MDD. However, it is important to note that, while most studies note a decrease in NE levels in MDD, LC activity in MDD, similar to PTSD, might still be increased.

The problem with dissecting the relationship between LC activity and NE release in clinical studies is a poor temporal resolution between chronic stress exposure, the clinical onset of the disorder, and biomarkers used to analyze pathology. Indeed, most studies assessing the levels of NE in MDD are carried out in postmortem studies, wherein the actions of stress within the LC–NE system might have been at their peak within the system. While seemingly counterintuitive, the decreased NE in MDD may be due to increased LC firing seen in stress. Studies demonstrate that increased or chronic stimulation of the LC in response to stress may lead to secondary depletion of NE as chronic stress increases CRH levels, which increases NE turnover, ultimately leading to NE depletion [48]. Indeed, postmortem brain analysis of suicide victims shows compensatory increases in tyrosine hydroxylase, the rate-limiting enzyme in NE synthesis, compared to normal controls [22]. In addition, released NE interacts with other neurotransmitter systems implicated in MDD, further complicating the analysis. Studies demonstrate that serotonergic neurons which release serotonin, another neurotransmitter altered in MDD, express $\alpha$2 receptors. Similar to noradrenergic neurons, $\alpha$2 receptor activation in serotonergic cells via NE administration inhibits cell firing [48]. Thus, any increases in NE may lead to decreased serotonin within the brain. Further studies must delineate the activity of LC neuronal firing and NE release within the context of MDD. Nevertheless, both anxiety and mood disorders demonstrate LC–NE system alterations. Taken together, these data suggest the involvement of stress-induced LC–NE dysregulation in the development of mood and anxiety disorders.

However, recent studies have demonstrated that inflammation is also dysregulated in these disease states, in addition to the HPA axis and LC–NE dysregulation.

## 4. The Effects of Chronic Stress on Peripheral Inflammation

A number of clinical studies provided early clues as to the implication of inflammation in the etiology of mood and anxiety disorders. In healthy control subjects, low-dose endotoxemia has been shown to increase peripheral inflammatory cytokines, which correlated with increased anxiety levels, depressed mood, and delayed memory recall [52]. Interestingly, administration of cytokine inhibitors in patients with MDD improves their depressive symptoms [53]. These findings suggest an inflammatory component to mood and anxiety disorders. Indeed, basal levels of peripheral pro-inflammatory cytokines are elevated in patients with depression and anxiety. In addition, increased peripheral levels of pro-inflammatory cytokines, IL-6, and c-reactive protein (CRP) were noted in MDD patients. PTSD patients, similar to MDD patients, also show increased pro-inflammatory cytokines [24], while increased CRP and TNFa levels were noted in patients with pathological anxiety [54]. Aside from increased peripheral pro-inflammatory cytokines, studies have also pointed out immune dysregulation [55] and expression of different regulatory T-cell (Treg) subsets in PTSD patients [56]. Unsurprisingly, increased inflammatory tone predisposes individuals to a PTSD diagnosis [57]. Collectively, these data allude to an inflammatory component of the etiology of mood and anxiety disorders. More recently, however, several studies have demonstrated a more direct link between stress and inflammation in mood and anxiety disorders.

Stress modulates and induces inflammation in mood and anxiety disorders through the action of excess GCs [58]. Typically, GCs suppress the inflammatory response in physiologic settings by preventing pro-inflammatory cell activation, downregulating pro-inflammatory cytokine production, and promoting healing. In addition, GCs inhibit inflammatory processes by repressing inflammatory transcription factors, inhibiting MAPK signaling, and promoting immune cell apoptosis [59]. Indeed, GCs are potent anti-inflammatory medications, used in the clinical setting for various inflammatory conditions; however, chronic stress exposure shift GCs from anti-inflammatory to pro-inflammatory [58]. Chronic stress promotes glucocorticoid-mediated TLR pathway activation, NLP3 induction, inflammasome formation in immune cells, and IL-6 release from endothelial cells [59]. Of particular interest, however, is the observation that, while injection of lipopolysaccharide (LPS), a component of Gram-negative bacteria and potent inducer of inflammation, in male mice increased TNFa, IL-1b, and IL-6, injection of LPS in mice that were also subjected to chronic restraint stress showed more dramatic increases in these measures, indicating that stress increased the effect of the inflammatory insult [60]. More interestingly, studies have demonstrated that stress can induce these pro-inflammatory changes without LPS-induced inflammation. Chronically-restrained male mice, without LPS injection, demonstrated decreased weight gain and increased peripheral levels of plasma CRP, TNFa, IL-6, and NE [61]. Together, these demonstrate the effects of stress in inducing a peripheral pro-inflammatory milieu. However, stress effects on inflammation are not limited to the periphery. Stress also induces neuroinflammation through microglia.

## 5. Microglia and the Induction of Neuroinflammation under Chronic Stress

Microglia, the primary inflammatory cell in the central nervous system, bridge the immune and neuronal systems. Microglia are motile cells that can respond to various peripheral inflammatory cues. Indeed, chief among microglia's many roles is immune surveillance. Through their motile processes, microglia continuously survey the cellular environment for potential insults [62]. In response to potential insults, microglia can phagocytose foreign material, release cytokines, and produce reactive oxygen species (ROS). Microglia can also recruit peripheral immune cells and other inflammatory effectors to the injury site through chemokine release [63]. Aside from immune surveillance, microglia are crucial to synaptic plasticity and neurodevelopment [64]. The overlapping microglial roles

of immune surveillance, synaptic plasticity, and neurodevelopment highlight the importance of microglia in both the immune and neuronal systems. Unsurprisingly, microglia exist in different morphologies and activation states to accomplish their many roles.

Microglial morphology ranges from ramified to ameboid and often correlates with microglial function [62]. Ramified microglia are considered "resting" and have motile, slender, branched processes that continually survey the cellular environment. On the other hand, ameboid microglia are considered "active" and have thicker, stout processes and round cell bodies [62]. It is important to note that, while there are distinct morphologies and activation states, microglia, due to their motility and dynamism, can exist in "in-between" states [65,66]. In addition to cell morphology and activation states, activated microglia can be further classified into phenotypes characterized by their "polarized" functions.

Activated microglia can be either pro-inflammatory (M1) or anti-inflammatory (M2), depending on the expression of key cellular markers and the release of specific cytokines and chemokines [67]. M1 microglia, characterized by cellular markers MHC-II and Iba-1, work to initiate inflammation and clearance of foreign bodies or pathogens. As such, M1 microglia release pro-inflammatory cytokines such as TNFa, IL-1b, IL-6, and IL-12 [68]. Also, M1 microglia release ROS and recruit peripheral inflammatory cells through the secretion of pro-inflammatory chemokines [69]. On the other hand, M2 microglia, characterized by cellular markers CD163 and CD206, contract the inflammatory response and promote healing [68]. In contrast to M1 microglia, M2 microglia release anti-inflammatory cytokines such as IL-4, IL-10, and IL-13 [67,70]. Thus, M1 and M2 homeostasis is critical in ensuring the inflammatory process clears the infection without damaging neurons. In addition to polarization, microglia can also assume a "primed" state.

Primed microglia are characterized by increased baseline expression of inflammatory markers and mediators, lower threshold for activation towards a pro-inflammatory state, and an exaggerated inflammatory response following immune activation [71]. Studies initially identified the phenomena of microglia in prion-infected mice, wherein the microglia of infected mice produced increased levels of IL-1b following immune challenge [72]. Similar primed microglial responses are noted with increased age, traumatic CNS injury, and neurodegenerative diseases [73]. Various secondary insults, such as stress exposure, trigger these primed microglia. Indeed, studies in mice demonstrate the potent effects of microglial priming and stress as a trigger. Early LPS injection in male mice increased neuronal spine engulfment in response to later stress exposure [74], indicating that inflammatory insult can affect neural plasticity through its actions on stress-induced reorganization of synaptic architecture. Because of the role in synaptic plasticity and morphological reorganization in brain systems that contribute to mood and anxiety disorders, this indicates that a history of elevated inflammation may increase risk for these types of diseases, and that pro-inflammatory signaling may represent a potential therapeutic target for their treatment. Indeed, mice injected with LPS early and exposed to stress later demonstrated increased susceptibility to depressive symptoms, indicating that microglial priming due to the inflammatory insult rendered the animals more susceptible to the subsequent stressor [74]. Likewise, concurrent LPS administration and chronic stress exposure in rats increase microglial density and decrease the proliferation of hippocampal stem cells [75]. In these studies, stress triggered the activation of primed microglia after the initial insult. More interesting, however, are studies indicating that chronic stress exposure alone can induce neuroinflammatory changes.

Pre-clinical studies demonstrate increased microglial recruitment and activation in response to various stressors. Indeed, mice subjected to chronic restraint stress without physical trauma or infections showed increased NE and peripheral pro-inflammatory cytokines [61]. Similarly, mice subjected to unpredictable chronic mild stress (UCMS) demonstrated increased hippocampal microglial density [76], which correlated with depressive-like behaviors [77]. Adding social isolation to UCMS also leads to anxiety-like and depressive-like behaviors and is associated with increased microglia without peripheral inflammation [78]. Pre-clinical PTSD models are also characterized by increased microglia

within the brain. Male mice subjected to the foot-shock model of PTSD demonstrated an increased proportion of brain microglia in the hippocampus, amygdala, and PFC [79]. These studies suggest the independent role of stress in inducing neuroinflammation. Unsurprisingly, chronic stress also induces polarization of activated microglia to the M1 phenotype.

In response to chronic stress, microglia increase the expression of M1-associated genes [80]. Consistent with M1 polarization, mice subjected to chronic stress showed elevated levels of pro-inflammatory cytokines TNFa and IL-6 [81] and recruitment of peripheral macrophages [80]. Stress-induced microglial activation also activates the NLRP1 inflammasome and decreases hippocampal BDNF [82], correlating with depressive-like behaviors in mice. Additionally, transcriptome profiling of male mice subjected to the resident intruder paradigm demonstrates differential time-dependent inflammatory gene expression in brain regions implicated in PTSD relative to the time of stress exposure. Expression of pro-inflammatory genes peaks early in the amygdala and is thought to be associated with fear maintenance, but later progresses towards the hippocampus and medial PFC, indicating a shift to memory consolidation and fear attenuation [83]. Likewise, anxiety- and depressive-like behaviors were observed in mice subjected to cumulative mild stress early in development, which correlated with increased microglia and regulatory T-cell 17 (Th-17) cells in the hippocampus and amygdala.

Interestingly, microglial inhibition has shown great promise in reversing anxiety- and depressive-like behaviors in pre-clinical models. Broad microglial inhibition with minocycline reversed chronic mild stress-induced depressive behaviors and reduced cytokine and NLPR3 cascade activation [84]. Similarly, minocycline administration rescued anxiety-like phenotypes [79]. Modulation of specific pro-inflammatory pathways has also shown positive pre-clinical effects on models of mood and anxiety disorders. After stress exposure, many pre-clinical studies have demonstrated increased NLRP1 [82] and NLRP3 [60,85] signaling. Increased inflammasome levels correlate with anxiety- and depressive-like phenotypes in rodents. Modulating the inflammasome pathways NLRP1 and NLPR3 had excellent outcomes. Inhibition of NLP3 cascade via administration of MCC950, a specific NLP3 inhibitor, attenuated anxiety-like behavior and prevents downstream transcription of IL-1b [86]. Likewise, knocking out the NLPR1 in mice stopped NLPR1-driven inflammation and rescued depressive-like behaviors [82]. Similarly, P2X7R antagonism, a purinergic receptor responsible for microglial activation, reduced hippocampal microglial density and rescued depressive-like behaviors in mice [87].

Modulation of downstream pro-inflammatory cytokine pathways such as IFNb and IL-1a also rescues anxiety- and depressive-like behaviors in pre-clinical models. Administration of IFN inhibitors to stressed mice restores social interaction and improves working memory compared to vehicle-treated mice [82]. Meanwhile, IL-1 receptor antagonist administration in rats exposed to witness stress rescued anhedonia [88], and IL-1R null mutant mice exhibited fewer anxiety-like behaviors compared to wild-type controls [89]. Interestingly, site-specific knockout of LC microglia through intra-LC clodrosome administration attenuated hypervigilant burying responses in stressed rats and attenuated accumulation of intra-LC IL-1b [88]. Furthermore, IL-17 administration reduced anxiety- and depressive-like behaviors, similar to SSRI administration, indicating that pro-inflammatory markers may represent important therapeutic targets for the treatment of symptoms of mood and anxiety disorders [90]. Together, these discoveries lend credence to exciting potential pathways for future research and drug treatments.

While the effects of stress and neuroinflammation are widespread throughout the central nervous system, the unique ability of LC to respond to both pro-stress and pro-inflammatory cues and signaling opens the possibility that it may represent a critical anatomic site where these types of signals are integrated to exert their cumulative effect.

## 6. Integration of Stress and Neuroinflammation in LC

Current studies implicate LC as an anatomic site that integrates stress signals and neuroinflammatory cues. As described previously, despite its small size, the LC responds to various brain regions and projects to different key brain regions [36]. This widespread projection allows the LC to modulate many central nervous system functions and behaviors. Indeed, LC responds to stress cues by increasing its tonic firing and NE release rates in its myriad terminal fields, leading to various behavioral changes, including state of arousal, altered responsiveness to sensory cues, and a generally hypervigilant behavioral state characterized by scanning labile attention [35]. However, in addition to stress, LC also responds to inflammatory cues. Administration of high-dose IL-1 in the LC increases tonic firing, similar to what occurs following stress or CRF administration [91]. Similarly, injection of IL-6 in male mice also increases LC activity, which correlates with anxiety- and depressive-like behaviors [92]. Studies have also shown that, in response to social intruder stress, male rats who were passive copers, measured by shorter latency to social defeat, showed higher LC *il-1* levels, while active copers showed the opposite [88]. Additionally, site-specific LC microglial depletion via clodronate administration attenuated hypervigilant responses in female rats in the context of witness stress, while also inhibiting IL-1b accumulation [93]. Aside from pro-inflammatory cytokines, decreases in anti-inflammatory cytokines were also noted in the LC after stress exposure. Acute exposure to immobilization stress decreased IL-4 while increasing anxiety-like phenotypes [94]. Collectively, these studies demonstrate LC's responsivity to inflammatory cues. However, LC can also modulate inflammatory cues.

In addition to inflammatory responsivity, LC, similar to GCs, modulates microglial dynamics and activation via NE release [62]. In many settings, NE acts as an anti-inflammatory agent. In vitro studies demonstrate that NE addition in microglial culture shortens microglial processes and slows motility [95]. Likewise, in vivo work in mice demonstrates increased microglial dynamics in anesthetized mice, wherein LC firing and NE release are the lowest relative to awake mice [96]. These studies suggest a negative correlation between NE and inflammation, wherein lower NE levels correlate with a more pro-inflammatory tone. Indeed, LC lesioning with DSP-4 increases levels of pro-inflammatory cytokines IL-1b, IL-2, IL-4, and TNFa in the dorsal raphe (DR) in rats with previous stress exposure [97]. NE released from the LC also takes part in the complex role of LC as a mediator of pro-stress and -inflammatory cues. While NE has anti-inflammatory roles, its effects are more complex, especially with the impact of dysregulated HPA axis and excess GCs. Increased NE has been shown to participate in pro-inflammatory priming within the background of HPA axis dysregulation and excess GCs. NE, acting through b-adrenergic receptors on microglia, can induce the production of pro-inflammatory cytokines. Pre-clinical studies demonstrate that the administration of a b-adrenergic agonist, isoproterenol, induces IL-1b production in male rats [98], while isolation of rat microglia treated with isoproterenol enhances the production of Il-1b and Il-6, suggesting a priming effect of NE in microglia [99]. Perhaps increased NE in mood and anxiety disorders primes microglia, which respond excessively to excess GCs, leading to a pro-inflammatory state. Indeed, increases in NE lead to a retracted or arrested state in microglia via microglia B2 receptors, leading to shorter outgrowth and decreased surveillance territory [100]. While decreased surveillance may be deemed beneficial, long-term decreases in surveillance can decrease critical microglia–neuronal interactions necessary for regulating neuronal activity, coordinating synaptic plasticity and maintaining brain homeostasis [101]. Hence, long-term decreased microglia–neuronal interactions can be detrimental. In addition, NE can also modulate adult neurogenesis. Studies have demonstrated that exogenous NE can impair adult neuroprogentor proliferation in the adult subventricular zone and reduce olfactory bulb neurogenesis [102]. These suggest that, apart from the inflammatory roles of microglia, their role in homeostasis is affected in response to chronic stress and NE release. Reductions in NE can increase microglial surveillance, promoting a pro-inflammatory tone [100]. However, as mentioned earlier,

chronic activation of the LC–NE system in mood and anxiety disorders can also lead to secondary reductions in NE.

Females are at higher risk for developing mood and anxiety disorders [3,103]. While many potential factors contribute to the higher risk observed in females, several possible mechanistic explanations have emerged, such as LC sexual dimorphism. Studies show that, in rats, females have larger LC compared to male rats [104]. Similarly, studies in humans reveal increased LC neurons in females compared to males [105,106]. The increased LC size and number in females suggest increased LC capacity and NE release in females compared to males. In addition to LC size and number, studies also show increased dendritic morphology in the female dorsolateral peri-LC area compared to male rats [104]. The dorsolateral peri-LC region receives afferents from limbic regions associated with mood and anxiety disorders [35]. Interestingly, increased dendritic morphology in the dorsolateral peri-LC could bias females to receive more CRF afferents from limbic regions, leading to higher LC activity in response to stress [104]. In addition to increased LC neuronal sensitivity and sensitization, female hormones such as estrogen exert effects on the LC–NE system. Estrogen increases NE synthesis and release, decreases NE degradation, and alters the expression of postsynaptic adrenergic receptors [104]. Perhaps, increased LC size and number and sustained NE in females contribute to increased sensitization in women. In response to stress, LC firing increases in females, leading to elevated NE in the periphery, priming the inflammatory system to release pro-inflammatory cytokines, which further increase LC firing, resulting in a feed-forward mechanism. Further studies must delineate the effect of sex and the contributions of NE and GCs within the context of mood and anxiety disorders. Nevertheless, these demonstrate a potential role for excess NE to prime microglia and induce inflammatory states.

Taken together, these studies demonstrate the unique role played by the LC as an integrator and responder to both stress and neuroinflammation. Such a role makes it critical in linking different complex etiologies in mood and anxiety disorders. The LC, despite its small size, provides an anatomic link between the complex effects of stress, the LC–NE system, and inflammation in mood and anxiety disorders. The relationships between these various risk factors, LC, and mood and anxiety disorders are summarized in Figure 1. Thus, through the LC, the complex intervening roles of each risk factor can be viewed with a coherent perspective. From this perspective, mood and anxiety disorders can be better understood, hopefully leading to better treatments.

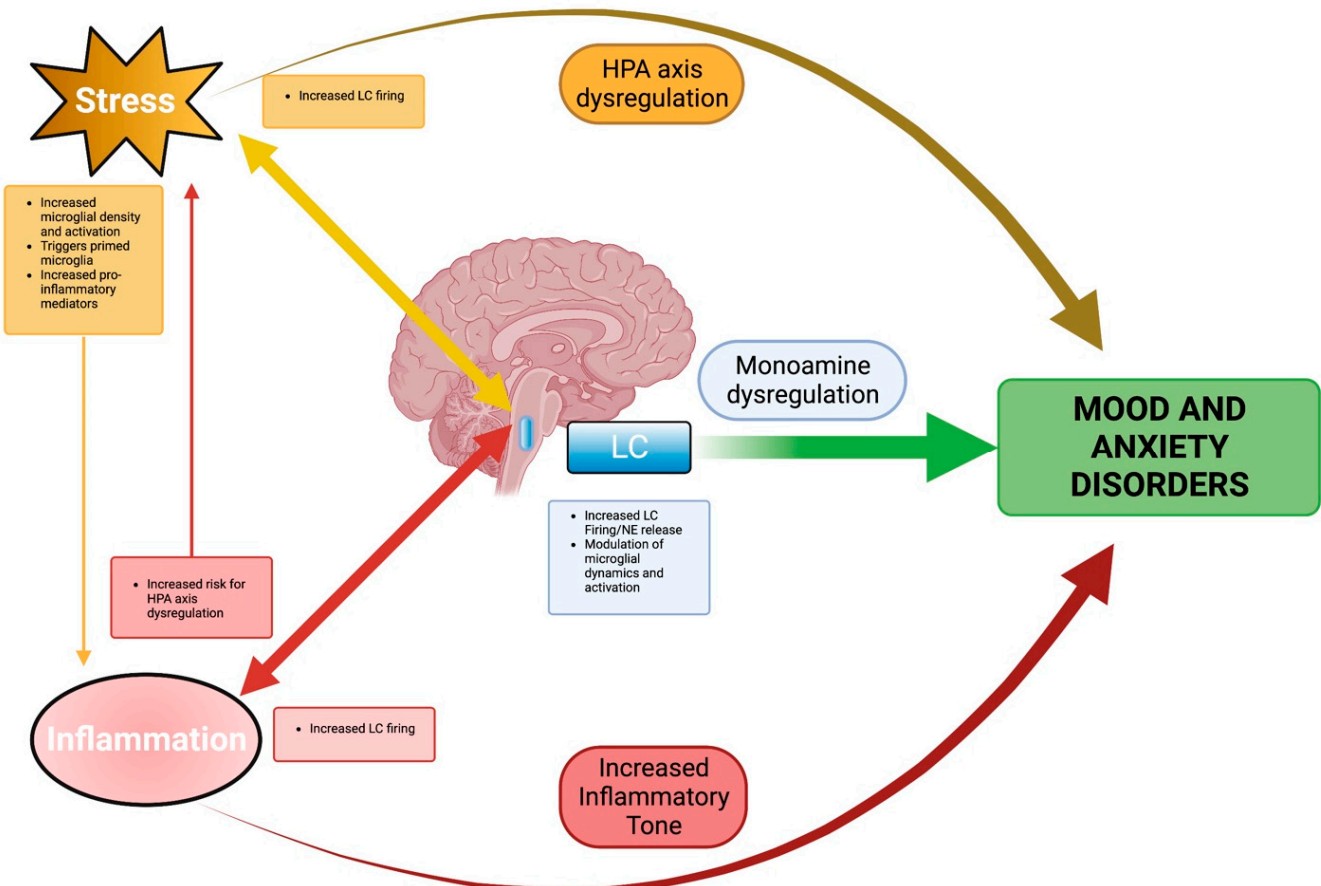

**Figure 1.** Summary Schematic. The locus coeruleus (LC) integrates stress and inflammation in mood and anxiety disorders. Stress influences inflammatory processes, both peripheral and central, toward pro-inflammatory shifts. Inflammation, meanwhile, creates an environmental milieu where stress responses, particularly glucocorticoid release, are potentiated. Independently, stress, through HPA axis dysregulation, and inflammation, through increased inflammatory tone, are identified as risk factors in mood and anxiety disorders. While common threads have been acknowledged, a link between these factors remains unclear. However, the reciprocal effects of stress and inflammation can be viewed through the LC. LC responds to and modulates stress and inflammation. Together, these modulating effects converge in LC, where monoamine dysregulation, stress, and inflammation contribute to the development of mood and anxiety disorders. Created with BioRender.

## 7. Conclusions

Mood and anxiety disorders are highly prevalent heterogeneous mental illnesses with complex etiologies. While the study of mood and anxiety disorders has identified key risk factors, treatment efficacies are still inadequate. Over the years, stress exposure has been identified as a key risk factor in mood and anxiety disorders. Stress affects multiple systems in the body to mount appropriate responses to stressors. However, constant stress exposure can lead to HPA axis dysregulation and precipitate mood and anxiety disorders. Indeed, patients with mood and anxiety disorders show HPA axis dysregulation. In addition, stress exposure also affects the LC–NE system and induces inflammatory changes, leading to increased mood and anxiety disorder susceptibility. The effects of stress, LC–NE dysfunction, and inflammation are complex and interrelated. Singling out each of their contributions is impossible. However, the effects of each can be better understood through the associated functions and dysfunctions of an integration site. Thus, an integrated perspective is critical to understanding the complex pathophysiology of mood and anxiety disorders. New studies point to the LC as a unique anatomic site integrating stress signals and inflammatory cues. Despite its minute size, studies reveal

the integration of the complex effects of stress, LC–NE dysfunction, and inflammation in the LC. Given its widespread connectivity, LC dysfunction due to chronic stress exposure, disease processes, or inflammatory changes can have broad-reaching effects. Further studies must identify LC projections under stress-induced inflammation and demonstrate underlying molecular processes driving these changes, to better elucidate LC's unique role. Nevertheless, an integrated perspective through the LC provides a valuable angle for viewing and understanding mood and anxiety disorders.

**Author Contributions:** A.A.A.R. and D.J.C. researched, wrote, and revised the manuscript. All authors have read and agreed to the published version of the manuscript.

**Funding:** This research was funded by New Jersey Health Foundation grant # PC-164-23, awarded to D.J.C.

**Institutional Review Board Statement:** Not applicable.

**Informed Consent Statement:** Not applicable.

**Data Availability Statement:** Not applicable.

**Conflicts of Interest:** The authors declare no conflict of interest.

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
