# Peer review of "Convergence of Pro-Stress and Pro-Inflammatory Signaling in the Central Noradrenergic System: Implications for Mood and Anxiety Disorders"

_2571-6980, doi:10.3390/neuroglia4020007_

Round 1

Reviewer 1 Report

Manuscript ID: 2350368

Title: Convergence of Pro-Stress and Pro-Inflammatory Signaling in the Central Noradrenergic System: Implications for Mood and Anxiety Disorders

Authors: Anthony A. Reyes and Daniel J. Chandler

This review article provides a nice summary of the how the locus coeruleus (LC) may mediate stress and inflammatory factors underlying mood disorders. The article is well-written and provides a clear summary of the stress and inflammatory factors linked to mood disorders, ending with a focus on the LC as a site positioned to integrate these mechanisms. The focus on the LC is well-reasoned and reflects an important area for the field. Overall, I found the review to be clearly written, interesting, and relevant. I have a few suggestions to improve the discussion about the role of the LC.

1) In addition to the human studies described that suggest elevated LC/NE signaling in PTSD patients, the authors should discuss clinical evidence for changes in NE in other mood disorders. Particularly with depression, in contrast to the PTSD relationship, several studies suggest reduced noradrenergic neurotransmission is associated with suicidal behavior (V. Arango, M.D. Underwood and J.J. Mann. (1996) Biol Psychiatry; G.A. Ordway, P.S. Widdowson, K.S. Smith and A. Halaris. J (1994) Neurochem), as well as MDD diagnosis (G.A. Ordway, J. Schenk, C.A. Stockmeier, W. May, V. Klimek. (2003) Biol Psychiatry, reported increase in agonist binding at α2-adrenergic autoreceptors on the cell body of NE neurons, indicating an increased function of these autoreceptors and therefore suggesting a decreased noradrenergic transmission in MDD). The authors should integrate these findings into their overall framework, as a reduced NE transmission seems to contradict the model laid out in the review (Figure 1 indicates increased LC firing/increased NE release).

2) Although the rationale for examining LC/NE mechanisms for mood disorders is clear, the final description of the model for how these systems may provoke mood disorders remains a little vague. The authors do a good job summarizing the effects that stress and inflammatory triggers have on NE release from LC (both increase NE). As the authors describe, however, NE itself often has anti-inflammatory properties. Their model (again, see Figure 1) indicates that elevated NE is related to mood disorders, in response to either stress or inflammation. Wouldn’t that increased NE be expected to reduce inflammation through its anti-inflammatory properties? If so, how would that would lead to a more depressed status? Clearly, the relationship among these factors is complex, and it would be helpful if the authors expanded this section to lay out potential dynamics. Perhaps chronic or developmental stress/immune challenge alters subsequent NE responses? Again, this may be tied into the fact that post-mortem human studies suggest reduced NE signaling (see above references).

3) Although the authors note that females are at higher risk for developing mood disorders, they do not provide any further discussion about potential mechanisms of this sex difference, particularly in regard to LC/NE. Given the growing push by the scientific community to consider sex differences, it would be helpful to include a brief discussion of what we know regarding sex differences in 1) the inflammatory consequences of stress (see Martinez-Muniz and Wood (2020), Journal of Pharmacology and Experimental Therapeutics) and the neuroanatomy of the LC (Bangasser DA, Wiersielis KR, and Khantsis S (2016) Brain Research). How would the observed sex differences in these systems factor into the model laid out in the review (higher susceptibility in women)?

Minor comment:

The authors may include the study by Kitayama, I., Otani, M., & Murase, S. (2004, Acta Neuropsychiatrica), as it provide evidence for increased microglia labeling in the LC in a rat model of depression.

Author Response

Reviewer 1
1.
We appreciate the valuable comment from the reviewer. We have included the human studies mentioned by the reviewer in lines 136-148 and proposed a mechanism accounting for decreased NE in MDD in our model in lines 150-162. In addition, we cited the recommended source by the reviewer (line 247, ref 76).

2. We acknowledge the reviewer's insightful comment regarding potential mechanisms of increased NE in the model. We realize that a statement regarding possible mechanisms is essential, and we have since added studies (Lines 319-335) expounding on our proposed models.

3. We have added the citations given by the reviewer and included a discussion on potential mechanisms of sex difference in our proposed model. (Lines 337-427).

Reviewer 2 Report

The review entitled “Convergence of Pro-Stress and Pro-Inflammatory Signaling in the Central Noradrenergic System: Implications for Mood and Anxiety Disorders” by Reyes et al, describes the effect of stress on mood and anxiety disorders. They pointed out that the central stress response, which drives behavioral and physiological change, is primarily mediated by activating the hypothalamic-pituitary-adrenal (HPA) axis, the locus coeruleus (LC), and by releasing the catecholamine transmitter norepinephrine (NE). The combined activity of the LC-NE system and HPA axis work synergistically to produce timely adaptive physiological and behavioral responses to stress. This brief review aims to summarize how LC is uniquely positioned to respond to both pro-stress and pro-inflammatory cues and how their convergence in this site may contribute to the development of mood and anxiety disorders. My comments are listed below.

Comments:

1.      Though various factors have been identified for the etiology of mood and anxiety disorders, why stress is important. Authors should include a brief overview of different risk factors and describe why stress is relatively important.

2.      Throughout the review-same statements are written many times. Authors should avoid this and try to describe it more logically. Simple flow charts can be helpful.

3.      The author has pointed out that there are microglia related pro-inflammatory and anti-inflammatory changes of cytokines and chemokines changes in stress disorder. Is there any specific pathway this activity follows? What are the future scopes for research in stress related mood and anxiety disorder?

4.      Is there any current medication that affects stress and may have some implications on mood and anxiety disorders.

Author Response

Reviewer 2
1.
We added a statement on line 64 regarding risk factors for mood and anxiety disorders and discussed the importance of stress in lines 71-72.

2. We thank the reviewer for this constructive comment. We have edited the document to reflect the reviewer's comments.

3. We recognize the reviewer’s point on including a discussion on altered pro-inflammatory pathways and future scopes for research. We have since added a section (Lines 264-302) discussing these changes.

4. We appreciate the insightful comment of the reviewer. However, a discussion of current medications that affect stress is outside the scope of the review. The study aims to propose LC as a convergence site between pro-stress and pro-inflammatory cues. We did include changes to discuss potential pathways for future research and drug discoveries in lines 264-302. (See response above)